# Using Heatmap Visualization to assess the performance of the DJ30 and NASDAQ100 Indices under diverse VMA trading rules

**Yuhsin Chen[1], Paoyu Huang[2], Min-Yuh Day[3], Yensen Ni[4]\*, Mei-Chu Liang[5]**

**1** Department of Accounting, Chung Yuan Christian University, Taoyuan, Taiwan, **2** Department of International Business, Soochow University, Taipei, Taiwan, **3** Graduate Institute of Information Management, National Taipei University, New Taipei, Taiwan, **4** Department of Management Sciences, Tamkang University, New Taipei, Taiwan, **5** Department of Banking and Finance, Tamkang University, New Taipei, Taiwan

\* ysniysni@gmail.com

**Data Availability Statement:** Weekly data are from Datastream for the DJ30 and NASDAQ100. The data is subject to confidentiality agreements with

## Abstract

We investigate whether using various VMA trading rules would improve investment performance due to the flexibility of VMA trading rules and the aid of Heatmap Visualization. Previously, investors frequently chose the best performance derived from limited VMA trading rules. However, our new design, which can display all results using Heatmap Visualization, shows that the NASDAQ100 index outperforms the DJ30 index and that weekly data outperforms daily data when measured by annualized return. These findings may be useful to those who trade index ETFs tracking the DJ30 and NASDAQ100 indices, as well as investors making investment decisions, and may contribute to the existing literature by evaluating the outcomes of VMA trading rules and providing insights for index ETF investors using a heatmap matrix, which is rarely explored and presented in the relevant literature.

## Introduction

Because inflation may erode the value of money deposited in banks, individuals may seek profit-generating investments in financial commodities to compensate for the loss of deposit value due to inflation [1,2]. As a result, many people may be motivated to invest in the stock market rather than deposit their money due to the fear of inflation. Additionally, profitability trading using technical trading rules appears to be welcomed by market participants and even academic researchers, which would be the research background and motivation for this study. We infer that the validity and usefulness of technical trading rules may exist because technical analysis indicators and technical analysis charts are disclosed on many prestigious financial websites around the world.

However, based on the market efficiency hypothesis (EMH) [3–5], there are three types of market efficiency: weak, semi-strong, and strong. That is, past stock price and volume information are fully reflected in stock prices in the weak form; all publicly available information, including insider information, is fully reflected in stock prices in the semi-strong form; and all information, including insider information, is fully reflected in stock prices in the strong form. However, relevant studies from various perspectives on stock market inefficiency, such as

third-party stakeholders, and cannot be shared openly due to intellectual property concerns. The contact information for Datastream is: https://www.refinitiv.com/en/products/datastream-macroeconomic-analysis. The authors did not receive any special privileges in accessing the data that other researchers would not have.

**Funding:** Min-Yuh Day has really appreciated the financial support provided by the Ministry of Science and Technology (MOST), Taiwan (110-2410-H-305-013-MY2), the National Taipei University (NTPU), Taiwan (111-NTPU_ORDA-F-001), and the National Taipei University (NTPU), Taiwan (111-NTPU_ORDA-F-003). The funders had no role in study design, data collection and analysis, decision to publish, or preparation of the manuscript.

**Competing interests:** The authors declare that they have no competing interests.

disposition effects [6,7], price overreaction [8–10], and even herding behaviors [11–13] appear to challenge such a viewpoint. Some investors, for example, may profit by using various technical indicators that can detect stock market overreaction [14,15]. These investors may employ contrarian strategies in response to stochastic oscillator indicator (SOI) and relevant strength indicator (RSI) signals of stock price overreaction [16,17]. Furthermore, other investors may profit by using momentum strategies as following-trend signals emitted by moving average (MA) trading rules that buy (sell or short sell) stocks when golden (dead) crosses occur [18]. In other words, these relevant studies may not support the EMH, implying that using technical trading rules properly is likely to generate profits.

Our motivation stems from the growing interest in the use of technical trading rules for profitability in the market and the need for a better understanding of the usefulness of technical trading rules in practice. We thus have emphasized the practical relevance of the study by highlighting the knowledge gap that still exists, despite the widespread adoption of technical trading rules in the market. Furthermore, because there are many different technical trading rules used in practice, we will focus on the MA trading rules because they are one of the most used trading rules [19–21].

Previous research has shown that using MA trading rules, such as the fixed-length moving average (FMA) and variable-length moving average (VMA), can result in profits [22–24]. According to Brock et al. [22], the most popular MA trading rule is 1–200 (i.e., using 1 day as the short-term MA (SMA) and 200 days as the long-term MA (LMA), and buy (sell) signals are emitted when the 1-day MA rises above (falls below) the 200-day MA). Furthermore, the VMA trading rules 1–50, 1–100, 1–150, and 1–200 are investigated in the relevant literature [23,25,26]. In terms of FMA trading rules, once a buying signal is generated using MA trading rules, it is held for a set period to estimate the holding period returns [27–29]. In other words, the FMA exit (i.e., the exit emitted after holding a certain fixed period) differs from the VMA exit (i.e., the exit is emitted after selling signals emitted by VMA trading rules).

Nevertheless, we stated that there is no consistent standard for setting a specific period as an exit signal in the FMA trading rules. In other words, the FMA trading rules will differ from the VMA trading rules in terms of exit signals, which define an explicit entry (exit) signal as the appearance of a golden cross (dead cross). Furthermore, Chang, Lima, and Tabak [30] demonstrate that the VMA trading rule has some forecasting power; Ratner and Leal [31] evidence that by employing several VMA trading rules in ten emerging equity markets of Latin America and Asia, VMA trading rules would be profitable for Taiwan, Thailand, and Mexico in particular; Ni, Lee, and Liao [32] clearly show that investors are likely to profit as buying signals emitted by VMA trading rules for Brazil, Furthermore, while evaluating the effectiveness of technical trading rules in cryptocurrency markets, Corbet et al. [24] reveal that VMA trading strategies would outperform other technical trading strategies in terms of profit generation. As a result, we argue that using VMA trading rules rather than FMA trading rules may be more appropriate.

As a result, we use VMA trading rules to evaluate the performance of these representative stock indices, such as the DJ30 and NSADAQ100 (NASDAQ100), because not only have these two indices received a lot of attention from the global media, but the trading volume of the constituent stocks of these indices is also very high. Furthermore, we discovered that index ETFs are very popular among investors because the size of index ETFs has increased in recent years. As a result, we believe that by using the DJ30 and NASDAQ100 indices as our investigated targets, our revealed results would provide a valuable reference for market participants investing in index ETFs (i.e., DJ30 ETFs and NASDAQ100 ETFs).

In practice, the 5, 20, and 60 present technical trading rules for a week, month, and quarter. VMA (5, 20) and VMA (5, 60) trading signals, for example, are emitted when the weekly MA

rises above (falls below) the monthly MA and when the weekly MA rises above (falls below) the quarterly MA. As a result, these MA trading rules are frequently used in practice, which differs from previous studies of MA trading rules such as 1–100 and 1–200 [22,23,25,26,32]. As a consequence, we included VMA (5, 20*N) trading rules, where N = 1 to 6, because VMA (5, 20) and VMA (5, 60) trading rules were enclosed in VMA (5, 20*N) trading rules. We then show that by using these representative indices, one of the VMA trading rules may outperform other trading rules. In the relevant literature, we argue that the research design can be considered the previous design [18,33].

Aside from the above design, we create a new design by utilizing the wisdom of heatmap visualization because a heatmap is a popular data visualization technique that is widely used in artificial intelligence and big data analytics to compare two-dimensional data as a matrix [34,35]. Van Craenendonck et al. [35], for example, stated that heatmap visualization techniques can help explain deep learning predictions in artificial intelligence image analysis and big data analytics [36]. Although there is a substantial amount of literature on heatmap data visualization in the field of computer science, there have been few studies that use heatmap data visualization techniques in the field of finance. Thus, in addition to the previous design described above, we propose a new design in which, by employing diverse SMA and LMA (i.e., VMA (n1, n2), where n1 is SMA, n2 is LMA, and n2>n1)), we can present numerous outcomes (i.e. geometric mean of stock index returns; GAs) in this heatmap matrix, representing the performance of employing diverse VMA (n1, n2) trading rules.

Based on our new design, we may obtain much higher results, even the highest result among numerous outcomes in Tables 3 and 4 (i.e., numerous outcomes derived from numerous VMA trading rules) or Figs 2 and 3 (i.e., numerous outcomes shown in many cells with different colors in a heatmap diagram). As such, we can select a better outcome area, even the best outcome among all of these numerous outcomes shown in a heatmap matrix from a bird's eye view, by employing numerous VMA trading rules due to the flexibility merit of VMA trading rules, which would be beneficial to market participants to select more appropriate VMA trading rules compared to the highest GA chosen by the previous design. As a result, we believe that this study may shed light on technical profitability trading from the standpoint of heatmap data visualization.

We argue that this study may contribute to the existing literature in several aspects. First, we attempt to determine whether investors would obtain better or even the best performance by evaluating numerous outcomes derived from employing numerous VMA trading rules for the data of these representative indices (i.e., DJ30 and NASDAQ100 indices). Second, we reveal the numerous outcomes (i.e. GAs) derived from employing VMA trading rules due to the flexibility merit of VMA trading rules, which appears to be rarely taken into account in the existing finance literature. Third, we may offer valuable insights for investors trading stock indices, particularly index ETFs. By utilizing a heatmap matrix, we provide a bird's eye view of numerous outcomes that can be used to screen for appropriate VMA trading rules based on different variable lengths (e.g., days or weeks) for SMA and LMA. We argue that our research design will be relevant to many investors trading index ETFs, as it enables them to potentially outperform those using the approach found in previous studies.

The rest of the paper is structured as follows. Section 2 describes the study's design and data. The empirical findings are presented in Section 3. Section 4 provides concluding remarks and recommendations.

## Design of this study

Because of the VMA trading rule used in this study, we first introduce the MA and then the VMA in SMA and LMA for various lengths (e.g., days or weeks).

## MA and VMA trading rules

The n-day MA is defined as follows:

$$MA_{t,n} = \frac{1}{n}\sum\nolimits_{i=t-n+1}^{t} P_i \qquad (1)$$

where $MA_{t,n}$ is the n-day moving average at time t, and $P_i$ is the closing price at time t.

According to the MA trading rule, the golden (dead) cross appears when the SMA rises above (falls below) the LMA, indicating the end of the downward (upward) trend and the beginning of the new upward (downward) trend. As a result, the golden and dead crosses are defined as follows.

$$\text{The golden cross is}: \ SMA_t > LMA_t \text{ and } SMA_{t-1} < LMA_{t-1} \qquad (2)$$

$$\text{The dead cross is}: \ SMA_t < LMA_t \text{ and } SMA_{t-1} > LMA_{t-1} \qquad (3)$$

Following MA trading rules, investors buy stocks when the golden cross appears and sell stocks when the dead cross appears, which is defined per trade. We examine the return (i.e., (selling price as the selling signal emitted by VMA / buying price as the buying signal emitted by VMA) -1) for each trade, and then measure the average return (AR) (i.e., the total sum of the return per trade / total trades) by evaluating each trade according to various VMA trading rules.

We would insert different lengths for SMA and LMA due to the flexibility of VMA trading rules, but the length of LMA would be longer than that of SMA. With the wisdom of selecting different lengths for SMA and LMA, we would create numerous combinations of VMA trading rules shown in the heatmap diagram, which would benefit investors to trade stocks and index futures by adopting appropriate lag lengths for SMA and LMA after observing a heatmap diagram from a bird's eye view.

## Research design

Since 5-day MA and 20-day MA are regarded as weekly MA and monthly MA in practice respectively, we then employ several VMA trading rules including VMA (5, 20*n) where n is from 1 to 6 in the beginning. However, by using daily data, we might not be able to generate profits or even suffer losses after taking transaction costs into account. Since we are concerned that long-term investments might outperform short-term investments, especially for the firms with increasing R&D [37,38], we thus adopt weekly data instead of daily data because the firms that put stress on R&D are those whose stocks are the constituent stocks of DJ30 and NASDAQ100 indices. We then employ various VMA trading rules including VMA (5, 20*n) where 5 indicates 5 weeks, 20 indicates 20 weeks, and n is from 1 to 6 as in the previous design of this study.

In addition, we must explain our new design. According to the general point of view, we should cover as many combinations as possible derived from various VMA trading rules. However, due to concerns about either avoiding limited trades (i.e., samples) that may skew our results or presenting all of the results in a heatmap matrix, we choose 30 days and 120 days as the maximum days for n1 and n2, with a five-week interval from 5 weeks to the maximum weeks of either n1 or n2 (i.e., the week set for SMA should be less than that set for LMA based on MA trading rules). Nonetheless, our new design incorporates all of the VMA trading rules from the previous design (i.e., VMA (5, 20*n), where n ranges from 1 to 6).

However, even though we have done our best to present our research results, the above concerns (i.e., avoiding limited samples and presenting all of the findings in a heatmap matrix)

may be one of the study's limitations because there appears to be no standard way of presenting our revealed outcomes. The first column in Tables 3 and 4 refers to n2 from 10 at the bottom to 120 at the top, and the last row refers to n1 from 5 at the leftmost to 30 at the rightmost. We would examine the performance of various VMA trading rules based on the combination of n1 and n2. As a result, we include not only all of the outcomes disclosed in the previous design but also present our overall outcomes in a heatmap data matrix, similar to a bird's eye view from heatmap visualization.

Following that, we can show our results in each table of Tables 3 and 4 (i.e., all of the results would be presented in a heatmap matrix (i.e., VMA (n1, n2), where n1 (i.e., short-term MA) is from 5 weeks to 30 weeks, n2>n1, and the interval is 5 weeks) or in each figure of Figs 2 and 3 (i.e., all of the results would be presented in a heatmap (i.e., all of the outcome derived from plentiful VMA would be displayed in a heatmap diagram).

## Measuring rate of return based on VMA trading rules

Based on VMA trading rules for trading financial commodities whose performances are closely related to these representative stock indices (e.g., index ETFs), we can calculate these index returns (i.e., IR) using the formula,

$$IRi = (Si/Bi) - 1 \tag{4}$$

where Si = closing index at the selling day (week) for trade i (i.e., the day (week) of selling signal emitted by a certain VMA trading rule)

Bi = closing index at the buying day (week) for trade i (i.e., the day (week) of buying signal emitted by a certain VMA trading rule)

The cumulative index return (i.e., CIR) is calculated using the formula below from IRi from k = 1 (first trade) to n (last trade).

$$CIR = \sum\nolimits_{k=1}^{n} (1 + IR_1)(1 + IR_2) \ldots \ldots \ldots \ldots (1 + IR_n) \tag{5}$$

where k = 1 to n (total trades produced using a certain VMA trading rule)

Following that, we can calculate the geometric average of these representative index returns (hereinafter referred to as GA), as shown below.

$$GA = \sqrt[n]{CIR} \tag{6}$$

where GA = geometric average of stock index returns

n = total trades produced using a certain VMA trading rule

In this study, after completing the first round-trip trade (buying the index ETFs as the first golden cross emitted by a certain VMA trading rule and selling index ETFs as the first dead cross emitted by a certain VMA trading rule), traders can proceed to the second round-trip trade (buying the index ETFs as the second golden cross emitted by the certain VMA trading rule and selling index ETFs as the second dead cross emitted by the certain VMA trading rule). Traders can process from the first trade (1) to the last trade (n) over the data period using the above trading process. As a result, we argue that measuring the geometric average would be more appropriate than measuring the arithmetic average in our research design because we can measure the next return after deriving the first return. Furthermore, we argue that investigating whether market participants would have higher GAs from a variety of VMA trading rules appears to be a rare concern in the existing literature since previous research has shown that by employing technical rules, they only present a limited number of results rather than a

**Table 1. Displays the means, standard deviations (SD), coefficient of variance (CV), median, minimum (Min), and maximum (Max) for DJ30 and NASDAQ100 indices based on weekly data from 2001 to 2020.**

|                | (1)    | (2)      | (3)     | (4)    | (5)      | (6)     | (7)      |
|----------------|--------|----------|---------|--------|----------|---------|----------|
| Stock index    | Sample | Mean     | SD      | CV     | Median   | Min     | Max      |
| DJ30           | 1044   | 14858.97 | 5880.39 | 39.57% | 12573.18 | 6626.94 | 30606.48 |
| NASDAQ100      | 1044   | 3374.12  | 2509.11 | 74.36% | 2213.19  | 815.40  | 12888.28 |

large number of results, demonstrating performance such as holding period returns, abnormal returns, and annualized returns [18,21,39,40].

Despite this, investors should consider transaction costs when trading index ETFs. The transaction cost of round-trip trading in index ETFs is at most 0.4% and often much lower (e.g., 0.1%). Since many GAs are over 5% shown in Tables 4 and 5, the transaction cost may not be the main issue in this study.

## Empirical results and analyses

### Descriptive statistics

Using weekly data from Datastream for the DJ30 and NASDAQ100 indices as our samples, Table 1 displays the mean, median, SD, CV, minimum, and maximum values for these indices from 2001 to 2020. The significant differences between the maximum and minimum values of these indices, as shown in Columns 6–7 of Table 1, indicate that the movement of these indices is quite volatile, as indicated by the standard deviation in Column (3). In addition, DJ30 and NASDAQ100 indices were plotted in Fig 1A and 1B. From 2001 to 2020, the trends of these indices are roughly upward, despite two notable declines (the 2008 stock market crash and the severe impact of COVID-19 at the start of 2020).

### Empirical results for previous research design

Because 5-day and 20-day MAs are deemed to be weekly and monthly MAs in practice, we employ VMA trading rules including VMA (5, 20*N), where N = 1–6 is the previous design, to address the concern of comparison. As a result, we begin with daily data and then examine the performances of these indices based on a variety of VMA trading rules (i.e., VMA (5, 20*N), where N = 1–6) as shown in Table 2.

However, while we treat these stock indices as financial instruments such as index ETFs, the returns of trading these financial instruments are rather disappointing for investors since the GA for trading such financial instruments is negative. Because of long-term investment recommended by institutional investors [41–44], we then use weekly data instead of daily data for further investigation according to our research design (i.e., we use the same research design except employing weekly data instead of daily data). The revealed results are shown in Table 3.

Table 3 shows that the performance of the NASDAQ100 using the VMA (5, 120) trading rule has the highest GA (60.17%), which is significantly higher than the GAs using other VMA trading rules (i.e., the rest of the GAs are all less than 30% in Table 3). In terms of the results for the DJ30 and NASDAQ100, we can see that the DJ30 performance is inferior to NASDAQ100 over the data period including daily data and weekly data, and the performance of using weekly data would be much better than that of using daily data, especially for using weekly data for NASDAQ100. Furthermore, we display information about these VMA trading rules such as the number of trades, the average duration day, and the maximum duration day. Although the highest GA is derived from trading the NASDAQ100 using the VMA (5,120) trading rule in the previous design, investors would be interested in obtaining higher GAs by

**Table 2. The results of using VMA trading rules for DJ30 and NASDAQ100 indices from 2001 to 2020 (daily data).**

| (1) | (2) | (3) | (4) | (5) | (6) |
|---|---|---|---|---|---|
| Strategy | No. of Trades | Total Return (%) | GA (%) | Avg. Duration Day | Max. Duration Day |
| Panel A: DJ30 | | | | | |
| (5, 20) | 319 | -65.00 | -0.20 | 16 | 89 |
| (5, 40) | 187 | -49.85 | -0.27 | 27 | 148 |
| (5, 60) | 153 | -29.74 | -0.19 | 32 | 245 |
| (5, 80) | 146 | -34.13 | -0.23 | 34 | 313 |
| (5, 100) | 124 | -37.61 | -0.30 | 40 | 341 |
| (5, 120) | 120 | -55.78 | -0.46 | 41 | 343 |
| Panel B: NASDAQ100 | | | | | |
| (5, 20) | 294 | -38.16 | -0.13 | 17 | 91 |
| (5, 40) | 199 | -51.60 | -0.26 | 25 | 127 |
| (5, 60) | 153 | -15.58 | -0.10 | 32 | 148 |
| (5, 80) | 117 | -7.06 | -0.06 | 42 | 324 |
| (5, 100) | 121 | -22.68 | -0.19 | 40 | 326 |
| (5, 120) | 101 | -8.91 | -0.09 | 48 | 439 |

using more VMA trading rules. Thus, we investigate whether different SMA and LMA combinations in a heatmap matrix yield higher GAs than the highest GA in the previous design in this study. Besides, even though the basis for using weekly data differs from that of using daily data, the results using weekly data are far superior to those using daily data when the annualized rate of return is measured.

## Empirical results for numerous outcomes shown in the heatmap visualization

Our new design is to generate the higher GAs among numerous GAs by employing numerous VMA trading rules, as shown in Tables 4 and 5 (i.e., the results of VMA (n1, n2) shown in a 6-by-23 heatmap matrix, where n1 (i.e., SMA) is from 5 weeks to 30 weeks, n2 (i.e., LMA) is

**Table 3. VMA trading strategies for DJ30 and NASDAQ100 from 2001 to 2020 (weekly data).**

| (1) | (2) | (3) | (4) | (5) | (6) |
|---|---|---|---|---|---|
| Strategy | No. of Trades | Total Return (%) | GA (%) | Avg. Duration Week | Max. Duration Week |
| Panel A: DJ30 | | | | | |
| (5, 20) | 62 | -14.66 | -0.24 | 16 | 70 |
| (5, 40) | 41 | -1.99 | -0.05 | 24 | 142 |
| (5, 60) | 25 | 16.67 | 0.67 | 39 | 191 |
| (5, 80) | 17 | -14.68 | -0.86 | 54 | 200 |
| (5, 100) | 17 | -13.75 | -0.81 | 54 | 203 |
| (5, 120) | 15 | 9.68 | 0.65 | 60 | 288 |
| Panel B: NASDAQ100 | | | | | |
| (5, 20) | 51 | 32.29 | 0.63 | 20 | 70 |
| (5, 40) | 41 | 73.09 | 1.78 | 24 | 138 |
| (5, 60) | 27 | 120.93 | 4.48 | 34 | 194 |
| (5, 80) | 17 | 310.55 | 18.27 | 54 | 338 |
| (5, 100) | 15 | 440.30 | 29.35 | 61 | 334 |
| (5, 120) | 9 | 541.51 | 60.17 | 100 | 585 |

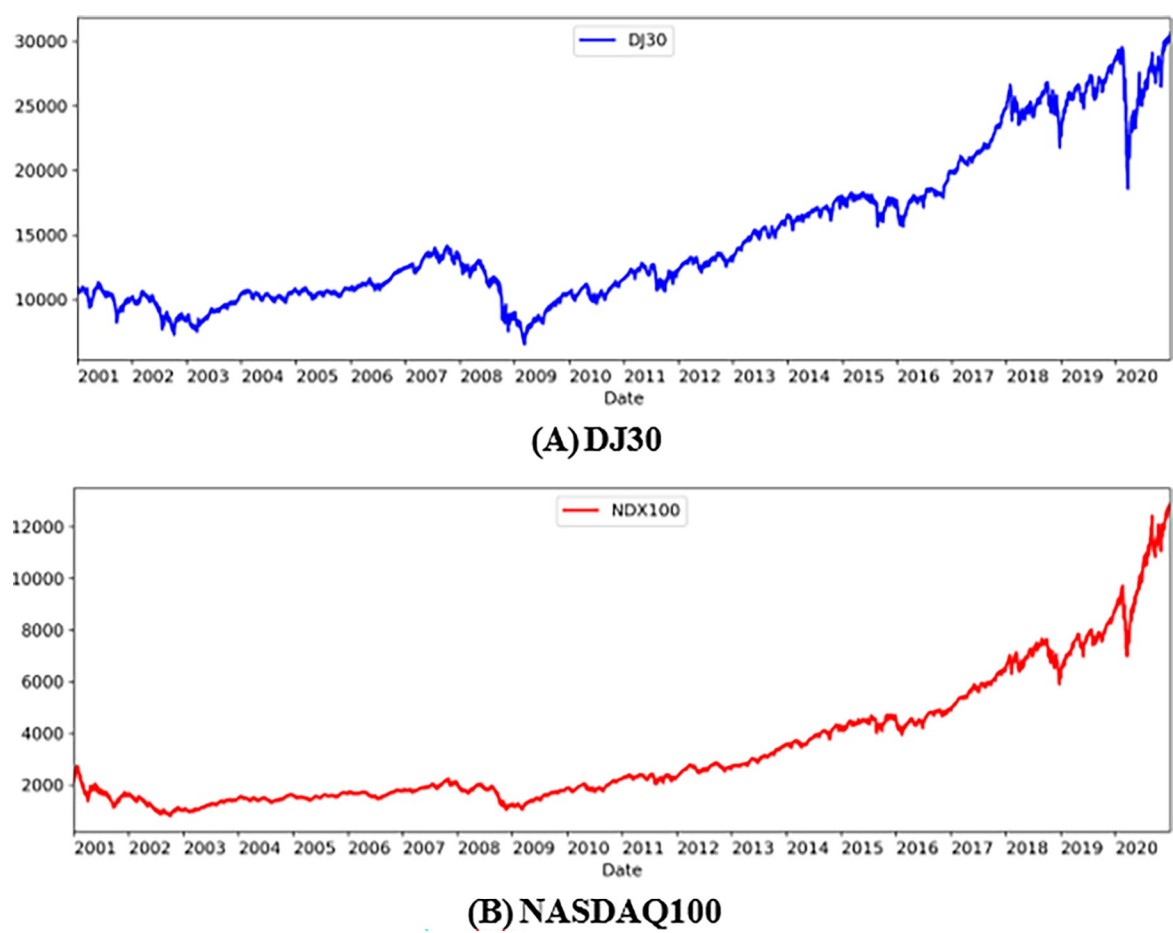

**Fig 1. The trend of the DJ30 and NASDAQ100 stock indices from 2001 to 2020.** (A) DJ30. (B) NASDAQ100.

from 10 weeks to 120 weeks, n2>n1, and the time interval is 5 weeks) or Figs 2 and 3 (i.e., the numerous GAs are depicted in a heatmap diagram with cells ranging from dark blue (representing much lower GAs) to bright yellow (representing much higher GAs)).

To gain a better understanding of the data processing results shown in this study's Tables 4 and 5 and Figs 2 and 3. The number shown in each cell in Tables 4 and 5 represents trading performance using diverse VMA (n1, n2) trading rules, and the cell with the number in bold represents trading performance (GM) that is greater than the highest GM using traditional trading rules. Furthermore, the color shown in each cell in Figs 2 and 3 indicated the performance using various VMA trading rules. The bright color represents better performance, which is useful for determining which area has better performance from a bird's eye view through heatmap visualization.

Following that, we would find bold cells representing higher GAs among numerous GAs in Tables 4 and 5 or bright color cells representing higher GAs from a bird's eye view via heatmap visualization in Figs 2 and 3 by employing different variable lags (weeks) for SMAs and LMAs. Because there are numerous outcomes in a heatmap matrix, we present the overall outcomes (GAs) in this heatmap matrix, which represent the performance of various VMA (n1, n2) trading rules. Table 4 shows, for example, that the performance (i.e. GA) of using VMA (5, 120) trading rules is 0.7%, which is equivalent to 0.65% rounding 0.7% (i.e., the GA of using the VMA(5, 120) trading rule) shown in Column (4) of Panel A in Table 3. However, we only

**Table 4. The DJ30 GA heatmap matrix using various round-turn trading rules.**

| 120 | *0.7* | 3.3 | 10.9 | 12.2 | 13.2 | 6.5 |
|---|---|---|---|---|---|---|
| 115 | 1.2 | **2.5** | 7.9 | 14.1 | 14.0 | 13.7 |
| 110 | 0.9 | **3.5** | 10.2 | 19.4 | 17.5 | 15.3 |
| 105 | -0.5 | **2.6** | 11.8 | 17.8 | 15.7 | 14.8 |
| 100 | ***-0.8*** | 3.1 | 11.1 | 20.4 | 17.4 | 16.1 |
| 95 | -0.9 | 3.8 | 9.1 | 19.4 | 16.1 | 13.9 |
| 90 | -0.2 | 2.7 | 8.7 | 17.5 | 16.6 | 15.4 |
| 85 | -1.1 | **2.2** | 12.4 | 16.6 | 17.2 | 14.5 |
| 80 | ***-0.9*** | 1.4 | 5.8 | 15.5 | 16.0 | 16.9 |
| 75 | -0.2 | 0.9 | **5.9** | 8.0 | 11.5 | 13.2 |
| 70 | -1.5 | **1.9** | 5.2 | 7.7 | 8.0 | 11.6 |
| 65 | -0.5 | 1.5 | 2.4 | 5.7 | 5.3 | 10.2 |
| 60 | *0.7* | 1.2 | 0.5 | 4.9 | 4.3 | 3.6 |
| 55 | 0.2 | -1.0 | -1.1 | 0.4 | **1.2** | -0.2 |
| 50 | -0.8 | -0.7 | -1.2 | **0.8** | 2.3 | 1.4 |
| 45 | -0.4 | -2.5 | -1.6 | 0.5 | 1.2 | 1.7 |
| 40 | ***-0.1*** | -1.2 | **1.1** | -1.2 | 0.4 | 0.1 |
| 35 | -0.1 | 0.1 | 0.2 | -0.9 | 1.0 | 0.2 |
| 30 | -0.8 | -0.3 | -0.8 | -0.4 | 0.7 | |
| 25 | -0.2 | -0.9 | -2.3 | -0.4 | | |
| 20 | ***-0.2*** | -0.1 | -1.8 | | | |
| 15 | -0.6 | -0.9 | | | | |
| 10 | -0.3 | | | | | |
| **n2 / n1** | **5** | **10** | **15** | **20** | **25** | **30** |

Note 1: Target: DJ30 index; Trading rule: VMA; Data: Weekly data from 2001 to 2020.

Note 2: n1 is from 5 weeks to 30 weeks, n2 is from 10 weeks to 120 weeks, n2>n1 (i.e., in bold), and the time interval is 5 weeks.

Note 3: The results of the previous design are shown in italic in Table 5, which are the same as the results shown in Column (4) of Panel A in Table 2. The results in bold cells are all higher than the highest GA of the previous design in Table 4.

observe 6 outcomes (i.e., GAs) in Column (4) of Panel A in Table 3 (i.e., the outcomes derived from the previous design), but we can observe numerous outcomes in our new design based on the combination of n1 and n2 employed for VMA (n1, n2) trading rules.

By displaying overall GAs derived by various VMA trading rules in Tables 4 and 5, we showed that using VMA trading rules, the GAs derived from trading the performance closely related to the NASDAQ100 (i.e., NASDAQ100 index ETFs) are significantly higher than those derived from trading the performance closely related to either the DJ30 (e.g., DJ30 index ETFs). Furthermore, in contrast to the previous design, which provided limited information for investors to exploit profits, the numerous GAs shown in Tables 4 and 5 (i.e., new design) would provide more valuable information to generate profits, even much higher profits, than the GAs shown in Panels A-B of Table 3 (i.e., previous design). For example, the DJ30 and NASDAQ100 results in bold cells in Tables 4 and 5 are significantly higher than the results in Panels A-B of Table 3.

In other words, when comparing the results of the new design to those of the previous design, our revealed results of the new design not only provide more information for investors

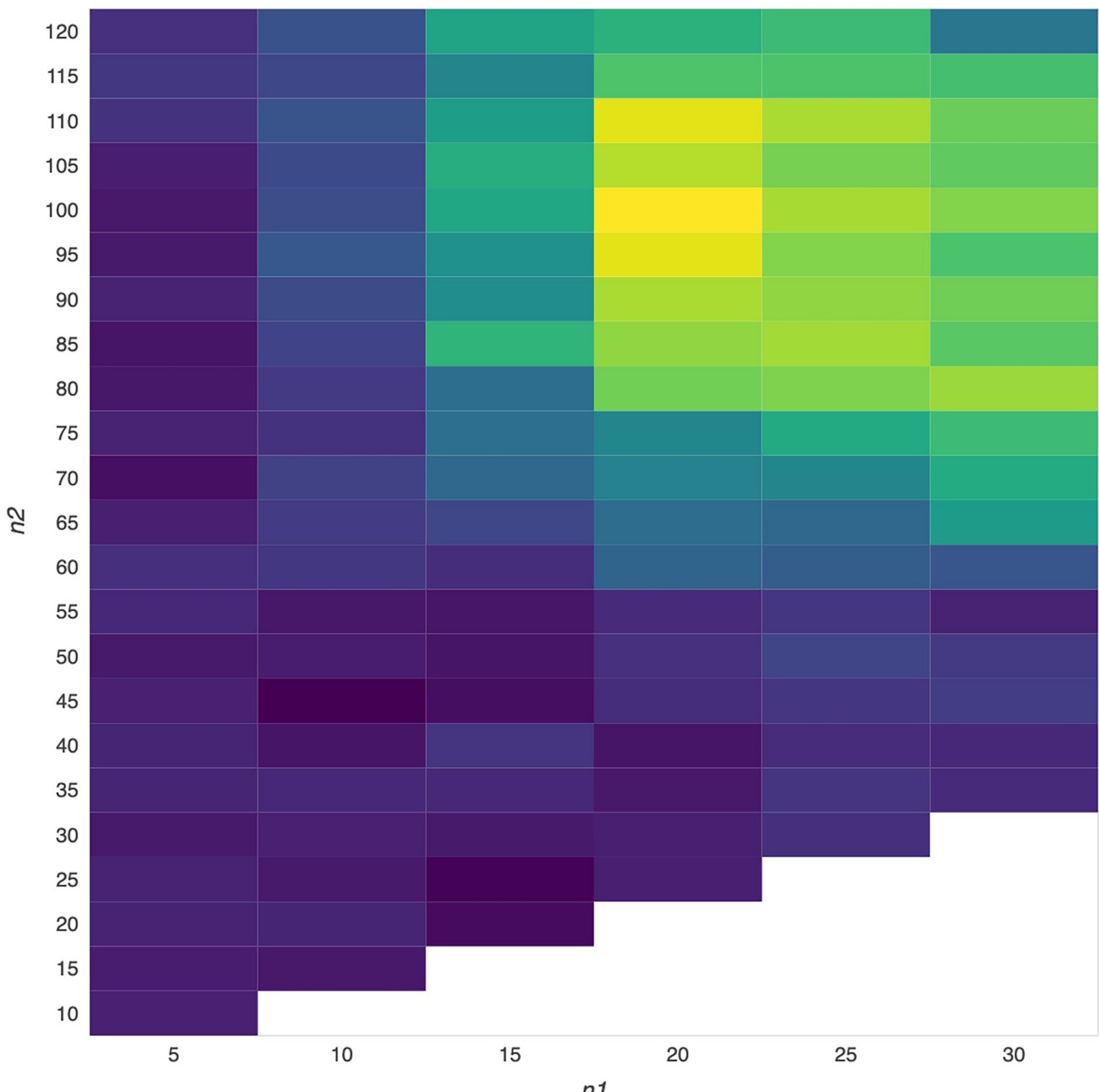

**Fig 2. Heatmap visualization of DJ30 trading results based on various VMA trading rules.**

to exploit higher profits in trading the performance closed related to these stock indices (i.e., these index ETFs), but our revealed GAs in a specific area (i.e. bold cells) are all significantly higher than the highest GA derived from the previous design. As a result, we argue that our new design is worthwhile for market participants trading the performance of index ETFs that are closely related to the DJ30 and NASDAQ 100 indices.

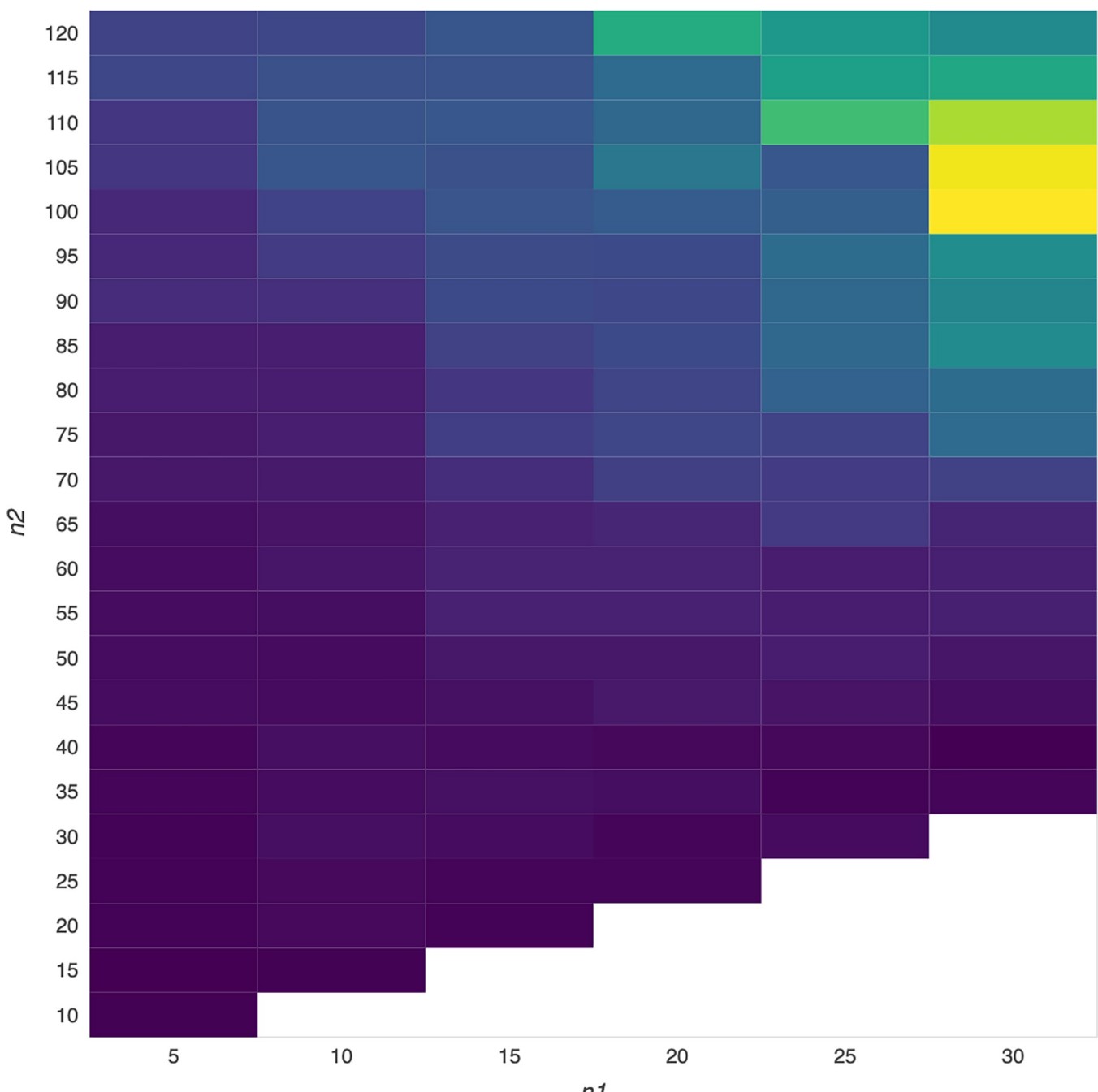

**Fig 3. Heatmap visualization of NASDAQ100 trading results based on various VMA trading rules.**

Despite this, investors trading these index ETFs should consider transaction costs; according to VMA trading rules, the transaction cost of round-trip trading for index ETFs is from 0.4% to 0.1%. According to the GAs shown in the heatmap (i.e., the results of the new design), most of the results are rather impressive. As a result, the transaction cost may be negligible. Furthermore, we may use a one-year treasury bill rate of around 1% as the benchmark because it represents the opportunity cost of using cash capital, and the average duration for round-trip trading using weekly data is nearly one year.

**Table 5. The NASDAQ100 GA heatmap matrix using various round-turn trading rules.**

| | | | | | | |
|---|---|---|---|---|---|---|
| 120 | *60.2* | 24.4 | 30.4 | **71.0** | **60.8** | 54.4 |
| 115 | 24.2 | 27.9 | 29.1 | 40.0 | **64.8** | **68.7** |
| 110 | 17.4 | 29.4 | 30.9 | 37.8 | **79.5** | **100.5** |
| 105 | 17.5 | 30.4 | 28.1 | 45.4 | 30.6 | **112.5** |
| 100 | *29.4* | 22.4 | 29.8 | 32.8 | 34.4 | **115.2** |
| 95 | 12.6 | 19.3 | 26.1 | 25.2 | 40.1 | 56.1 |
| 90 | 13.8 | 15.2 | 25.4 | 24.3 | 37.9 | 52.0 |
| 85 | 8.5 | 9.0 | 21.6 | 25.3 | 37.8 | 55.0 |
| 80 | *18.3* | 8.1 | 17.7 | 23.4 | 35.8 | 40.3 |
| 75 | 6.6 | 9.1 | 20.8 | 24.1 | 23.1 | 39.6 |
| 70 | 6.5 | 7.8 | 14.5 | 21.2 | 19.2 | 22.3 |
| 65 | 3.4 | 5.3 | 9.9 | 11.6 | 18.9 | 11.0 |
| 60 | *4.5* | 6.2 | 10.7 | 10.4 | 7.9 | 9.4 |
| 55 | 3.0 | 3.7 | 9.7 | 9.7 | 7.9 | 9.3 |
| 50 | 2.9 | 2.6 | 6.5 | 6.8 | 8.6 | 6.3 |
| 45 | 2.9 | 2.5 | 4.5 | 7.3 | 5.4 | 3.6 |
| 40 | *1.8* | 4.1 | 2.8 | 1.6 | 1.6 | -0.8 |
| 35 | 0.9 | 3.1 | 4.5 | 3.3 | 0.3 | 1.2 |
| 30 | 0.4 | 4.0 | 3.3 | 1.4 | 2.6 | |
| 25 | 0.3 | 2.3 | 0.8 | 1.1 | | |
| 20 | *0.6* | 2.3 | 0.5 | | | |
| 15 | -0.3 | 0.0 | | | | |
| 10 | -0.1 | | | | | |
| n2 / n1 | 5 | 10 | 15 | 20 | 25 | 30 |

Note 1: Target: NASDAQ100; Trading rule: VMA; Data: Weekly data from 2001 to 2020.

Note 2: n1 is from 5 weeks to 30 weeks, n2 is from 10 weeks to 120 weeks, n2>n1 (i.e. in bold), and the time interval is 5 weeks.

Note 3: The results of the previous design are shown in italic in Table 5, which are the same as the results shown in Column (4) of Panel B in Table 2. The results in bold cells are all higher than the highest GA of the previous design in Table 5.

## Discussion

Based on our findings, our discussion is as follows. First, our study aimed to determine if investors could achieve better performance by evaluating various outcomes derived from employing different VMA trading rules for the DJ30 and NASDAQ100 indices. This is important because it addresses a crucial question that many investors have: which trading rules should they use to achieve optimal performance? By providing empirical evidence on the performance of different VMA trading rules, our study offers valuable insights for investors looking to enhance their trading strategies. Second, our study reveals the numerous outcomes (i.e., GAs) derived from employing VMA trading rules due to their flexibility. This highlights the need for investors to consider the flexibility of trading rules when selecting the appropriate ones. This contribution is valuable to the finance literature as it provides a more nuanced understanding of VMA trading rules' performance. Third, our study provides insights for investors trading stock indices, particularly index ETFs. We achieve this by utilizing a heatmap matrix to provide a bird's eye view of numerous outcomes that can be used to screen for appropriate VMA trading rules based on different variable lengths for SMA and LMA. This aspect is significant as it offers a practical tool for investors to enhance their trading strategies.

We believe this contribution is particularly appealing to readers as it offers a clear and concise way to analyze and select appropriate VMA trading rules. Overall, our research design will be relevant to many investors trading index ETFs, as it enables them to potentially outperform those using the approach found in previous studies. By providing empirical evidence, highlighting the flexibility of trading rules, and providing practical tools for investors, our study is a valuable addition to the finance literature.

## Concluding remarks

Since trading financial instruments (e.g., stocks and bonds) based on technical trading indicators are widely used in many well-known financial websites such as Bloomberg, Market Watch, and Forbes, we intend to investigate whether investors would generate higher profits by employing numerous VMA trading rules due to the flexibility merited by employing different lag lengths for SMA and LMA for VMA trading rules, which appear to be rarely addressed in the existing literature. As a result, we argue that the effectiveness of the VMA trading rule is worth investigating. As history often repeats itself, many investors may generate profits and even more profits by referencing our revealed results derived from numerous VMA trading rules by analyzing historical data, including long-term data.

In this study, we attempt to obtain the higher GA area and even the highest GA, among numerous GAs shown in a heatmap diagram from a bird's eye view [45,46], which may make it easier and more beneficial for investors to select the appropriate VMA trading rule for generating profit exploitation. We argue that our heatmap design is superior to the previous design because the GAs shown in a specific area of the heatmap diagram are all higher than the highest GA derived from the previous design, which would rather impress investors who are interested in trading the performance closely related to the DJ30 and NASDAQ100 indices (i.e., index ETFs of DJ30 and NASDAQ100).

Furthermore, this research could have two important practical implications. First, we argue that based on the notion that a more systematic interaction of theory and practice would be more valuable [20], investors may be able to generate higher returns if they can measure a variety of outcomes using various VMA trading rules, present these outcomes in a heatmap diagram from a bird's eye view, classify these outcomes, and then use appropriate VMA trading rules to trade these ETFs and other financial instruments. Second, we argue that well-preparation would be the prerequisite for enhancing profitability and even reducing risks in trading these index ETFs, especially these index ETFs held by so many investors and even global investors, and thus this study may provide valuable information for investors to enhance profitability in trading these index ETFs.

Moreover, we would discuss the study's strengths and weaknesses. Concerning the study's strengths, we argue that it is linked to representative US stock indices (i.e., DJ30 and NASDAQ100 indices), investing strategies (i.e., momentum strategies implemented by VMA trading rules), and screening trading rules (i.e., heatmap visualization beneficial for selecting proper trading rules), which may contribute to the existing literature because our research not only addresses the concerns raised above but also provides useful information to investors who trade these index ETFs. Regarding the study's possible flaws, even though we set numerous VMA trading rules, these still have more than numerous VMA trading rules if we extend our variable lag length or set a short interval.

### Limitations and further research

In this study, the sample size would be limited for several VMA trading rules (i.e., VMA (5, 120)) even though we use long-period data (i.e., 20-year data). As such, we should point out

that limited trading signals generated by several VMA trading rules would be the limitation of this study. Besides, based on the limitation of this study, we argue that future research and potential research extensions for this study should be concerned with the following. First, due to using weekly data, there would be a fact that the samples size would be limited for several VMA trading rules, which would be the limitation of this study, As such, we might employ the constituent stocks of NDX 100 and S&P 500 indices for further research, which should have enough samples for diverse VMA trading rules, which would provide more reliable results in our future studies. Second, because our heatmap is static, it may contain limited information and be of limited use. As such, providing a signaling approach that uses dynamic heatmaps and heatmap-aided dynamic VMA rules as future research may be much better and even beat the single/static VMA rules, thereby contributing to the existing literature. Third, we would extend our variable lag lengths and shorten our intervals for more VMA trading rules, compare the findings in other financial markets to those in this study, and extend our research design to other financial instruments (e.g., index futures) and even commodity instruments (e.g., gold and oil), which may generate more profitable opportunities than the previous design or other research designs.

## Author Contributions

**Conceptualization:** Yuhsin Chen, Paoyu Huang, Min-Yuh Day, Yensen Ni.

**Data curation:** Mei-Chu Liang.

**Methodology:** Yuhsin Chen, Paoyu Huang, Min-Yuh Day, Yensen Ni.

**Software:** Yuhsin Chen, Paoyu Huang, Min-Yuh Day, Yensen Ni.

**Validation:** Mei-Chu Liang.

**Writing – original draft:** Yuhsin Chen, Paoyu Huang, Min-Yuh Day, Yensen Ni, Mei-Chu Liang.

**Writing – review & editing:** Yuhsin Chen, Paoyu Huang, Min-Yuh Day, Yensen Ni, Mei-Chu Liang.

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
