## [Decision Letter · Decision Letter 0]

20 Jan 2023

PONE-D-22-31748Using Heatmap Visualization to assess the performance of the DJ30 and NASDAQ100 Indices under diverse VMA trading rulesPLOS ONE

Dear Dr. Ni,

Thank you for submitting your manuscript to PLOS ONE. After careful consideration, we feel that it has merit but does not fully meet PLOS ONE’s publication criteria as it currently stands. Therefore, we invite you to submit a revised version of the manuscript that addresses the points raised during the review process.

The manuscript requires further revisions with reference to prior literature discussion, empirical methodology, outcomes’ interpretation, as well as policy implications. As well, the contribution of research to the existing literature should be deeply argued.

We look forward to receiving your revised manuscript.

Kind regards,

Stefan Cristian Gherghina, PhD. Habil.

Academic Editor

PLOS ONE

Journal Requirements:

"Min-Yuh Day has really appreciated the financial support provided by the Ministry of Science and Technology (MOST), Taiwan (110-2410-H-305-013-MY2), the National Taipei University (NTPU), Taiwan (111-NTPU_ORDA-F-001), and the National Taipei University (NTPU), Taiwan (111-NTPU_ORDA-F-003)."

4. Please upload a copy of Figure 2, to which you refer in your text on pages 6 and 10. If the figure is no longer to be included as part of the submission please remove all reference to it within the text.

Reviewers' comments:

Reviewer's Responses to Questions

**Comments to the Author**

1. Is the manuscript technically sound, and do the data support the conclusions?

Reviewer #1: Yes

Reviewer #2: No

Reviewer #3: Yes

2. Has the statistical analysis been performed appropriately and rigorously? 

Reviewer #1: Yes

Reviewer #2: No

Reviewer #3: Yes

3. Have the authors made all data underlying the findings in their manuscript fully available?

Reviewer #1: Yes

Reviewer #2: Yes

Reviewer #3: Yes

4. Is the manuscript presented in an intelligible fashion and written in standard English?

Reviewer #1: Yes

Reviewer #2: Yes

Reviewer #3: Yes

5. Review Comments to the Author

Reviewer #1: Please find the attached file in my attachments to see my detailed comments. In this report, I would like to point some points which might be helpful for the authors to improve the quality of the manuscript before it gets published.

Reviewer #2: The authors explore various VMA (variable-length moving average) trading rules and propose to use heatmap visualization to display the results. The authors use Dow Jones 30 and NASDAQ 100 as subjects of investigation and find greater return when using weekly data. The VMA trading rule is in general based on a long-term and a short-term moving average of an index, and once short-term MA is above (or below) long-term MA, it indicates a signal to buy (or sell) the index-linked product (see p.8 as well as Eq (2) & (3) in the manuscript). Although the Dow Jones and NASDAQ indexes are not really tradable, the authors would like to shed light on tracking the performance of exchange traded funds linked to these index as well as investment decision-making.

There are major and minor concerns relating to the interpretations and the contributions. Please find my detailed concerns in the attached report.

Reviewer #3: This is an interesting topic. The methods is suitable for this research purpose. The research findings are good for investors' decision.

There are some comments for the authors as follows:

1. The author should provide more academic contribution of this research in the abstract.

2. The authors should discuss more details of the data processing results

3. The discussion and conclusion should be written separately so the authors can have many space for discussion.

6. PLOS authors have the option to publish the peer review history of their article (what does this mean?). If published, this will include your full peer review and any attached files.

Reviewer #1: **Yes: **Toan Luu Duc Huynh

Reviewer #2: No

Reviewer #3: No

---

## [Author Response · Author response to Decision Letter 0]

30 Mar 2023

Please see Responses to Reviewers, and thank you very much!

---

## [Decision Letter · Decision Letter 1]

12 Apr 2023

Using Heatmap Visualization to assess the performance of the DJ30 and NASDAQ100 Indices under diverse VMA trading rules

PONE-D-22-31748R1

Dear Dr. Ni,

We’re pleased to inform you that your manuscript has been judged scientifically suitable for publication and will be formally accepted for publication once it meets all outstanding technical requirements.

Kind regards,

Stefan Cristian Gherghina, PhD. Habil.

Academic Editor

PLOS ONE

Additional Editor Comments (optional):

Reviewers' comments:

Reviewer's Responses to Questions

**Comments to the Author**

1. If the authors have adequately addressed your comments raised in a previous round of review and you feel that this manuscript is now acceptable for publication, you may indicate that here to bypass the “Comments to the Author” section, enter your conflict of interest statement in the “Confidential to Editor” section, and submit your "Accept" recommendation.

Reviewer #1: All comments have been addressed

Reviewer #2: All comments have been addressed

2. Is the manuscript technically sound, and do the data support the conclusions?

Reviewer #1: Yes

Reviewer #2: Yes

3. Has the statistical analysis been performed appropriately and rigorously? 

Reviewer #1: Yes

Reviewer #2: (No Response)

4. Have the authors made all data underlying the findings in their manuscript fully available?

Reviewer #1: Yes

Reviewer #2: (No Response)

5. Is the manuscript presented in an intelligible fashion and written in standard English?

Reviewer #1: Yes

Reviewer #2: Yes

6. Review Comments to the Author

Reviewer #1: The study has addressed all previous comments; therefore, I do not have any further comments on this manuscript and I am happy to accept this paper.

Reviewer #2: (No Response)

7. PLOS authors have the option to publish the peer review history of their article (what does this mean?). If published, this will include your full peer review and any attached files.

Reviewer #1: No

Reviewer #2: No

---

## [Editor Report · Acceptance letter]

4 May 2023

PONE-D-22-31748R1 

Using Heatmap Visualization to assess the performance of the DJ30 and NASDAQ100 Indices under diverse VMA trading rules 

Dear Dr. Ni:

I'm pleased to inform you that your manuscript has been deemed suitable for publication in PLOS ONE. Congratulations! Your manuscript is now with our production department. 

Kind regards, 

on behalf of

Dr. Stefan Cristian Gherghina 

Academic Editor

PLOS ONE